

# Effects of changes in isotopic baselines on the evaluation of food web structure using isotopic functional indices

Simon Belle[1,2] and  Gilbert Cabana[2]

[1] Department of Aquatic Sciences and Assessment, Swedish University of Agricultural Sciences, Uppsala, Sweden

[2] Centre de Recherche sur les Interactions Bassins Versants-Ecosystèmes Aquatiques (RIVE), Université du Québec à Trois-Rivières, Trois-Rivières, Canada

## ABSTRACT

**Background**. This study aimed to assess whether ecological inferences from isotopic functional indices (IFIs) are impacted by changes in isotopic baselines in aquatic food webs. We used sudden $CO_2$-outgassing and associated shifts in DIC-$\delta^{13}C$ brought by waterfalls as an excellent natural experimental set-up to quantify impacts of changes in algal isotopic baselines on ecological inferences from IFIs.

**Methods**. Carbon ($\delta^{13}C$) and nitrogen ($\delta^{15}N$) stable isotopic ratios of invertebrate communities sharing similar structure were measured at above- and below-waterfall sampling sites from five rivers and streams in Southern Quebec (Canada). For each sampled invertebrate community, the six Laymans IFIs were then calculated in the $\delta$-space ($\delta^{13}C$ vs. $\delta^{15}N$).

**Results**. As expected, isotopic functional richness indices, measuring the overall extent of community trophic space, were strongly sensitive to changes in isotopic baselines unlike other IFIs. Indeed, other IFIs were calculated based on the distribution of species within $\delta$-space and were not strongly impacted by changes in the vertical or horizontal distribution of specimens in the $\delta$-space. Our results highlighted that IFIs exhibited different sensitivities to changes in isotopic baselines, leading to potential misinterpretations of IFIs in river studies where isotopic baselines generally show high temporal and spatial variabilities. The identification of isotopic baselines and their associated variability, and the use of independent trophic tracers to identify the actual energy pathways through food webs must be a prerequisite to IFIs-based studies to strengthen the reliability of ecological inferences of food web structural properties.

Corresponding author
Simon Belle, simon.belle@slu.se

## INTRODUCTION

Stable isotopes analysis, mainly those of carbon and nitrogen, of aquatic consumers is a common technique to provide quantitative and qualitative measurements of energy flows in food webs (*Cabana & Rasmussen, 1996*; *Post, 2002*; *Vander Zanden et al., 2016*). Consumer isotopic ratios are often represented in a $\delta$-space (i.e., $\delta^{13}C$-$\delta^{15}N$ biplot), where species trophic interactions can be assessed using a large variety of analytical tools (*Layman et al., 2012*). Among them, isotopic functional indices (IFIs) are based on the distribution

and the dispersion of species in $\delta$-space and have been developed to calculate measures of trophic structure of food webs (*Layman et al., 2007*; *Jackson et al., 2011*; *Cucherousset & Villéger, 2015*). Briefly, IFIs allow to infer food web structural properties, and can be grouped according to three major components of trophic diversity. First, isotopic functional richness providing a quantitative indication of the extent of isotopic space of the entire community (*Layman et al., 2007*; *Jackson et al., 2011*). Secondly, isotopic functional divergence providing information on the average degree of trophic diversity within a $\delta$-space (*Layman et al., 2007*), and thirdly, isotopic functional evenness quantifies the regularity in species distribution and may also be seen as an indicator of trophic redundancy in food webs (*Layman et al., 2007*; *Rigolet et al., 2015*).

The IFI concept is, however, based on two main assumptions: that two close species have similar role in food webs; and that isotopic metrics are good proxies of food web structural properties (*Layman et al., 2007*), but too few empirical studies have been conducted to evaluate the validity of these underlying assumptions (*Syväranta et al., 2013*; *Jabot et al., 2017*). Several authors have, however, pointed out that overlaps and variabilities in isotopic baselines could be major pitfalls of IFIs and hamper identification of actual food web structure (*Hoeinghaus & Zeug, 2008*; *Jabot et al., 2017*), but very few studies have empirically tested for the sensitivity of IFIs to these issues (*Jackson et al., 2011*). Moreover, differences in IFIs sensitivities to changes in isotopic baselines could be inherently driven by differences in calculation methods: being higher for IFIs based on dispersion of species in the $\delta$-space than for others based on their distribution. For instance, several authors have suggested that isotopic functional richness indices (i.e., measuring species dispersion in the $\delta$-space) are strongly influenced by changes in ranges of consumers $\delta^{13}$C and $\delta^{15}$N values (*Brind'Amour & Dubois, 2013*; *Syväranta et al., 2013*), and ecological inferences of food web structural properties from these scale-dependent IFIs are therefore highly sensitive to changes in isotopic baselines.

Carbon of aquatic consumers sustained by autochthonous food resources (i.e., algae) is derived from the fixation by autochthonous primary producers of dissolved inorganic carbon (DIC) during photosynthesis. In river ecosystems, many biological and biochemical processes (i.e., respiration, water flow velocity, etc.) can influence DIC- $\delta^{13}$C values (see also *Finlay, 2003*), leading to strong spatial/temporal variability in algal $\delta^{13}$C values (*France & Cattaneo, 1998*; *Finlay, 2001*; *Rasmussen, 2010*). Due to this large variability in isotopic baseline over time/space, different diets could lead to similar isotopic ratios of aquatic consumers, and conversely same diets could have different isotopic ratios. Changes in isotopic ratios of basal food resources can thus lead to potential misinterpretations of IFIs in river studies comparing food webs across sites and/or over time, and complementary empirical studies are needed to better assess whether ecological inferences from IFIs are impacted by variabilities in isotopic baselines.

Artificial variability in river algal $\delta^{13}$C values can be acquired by manipulating DIC-$\delta^{13}$C values (*Cole et al., 2002*). In that vein, artificial tracer studies (i.e., $^{13}$C-tracer addition experiments) have been conducted in small streams to induce changes in isotopic baselines of algal food resources and track the fate of algal biomass in stream food webs (*Hotchkiss &*

*Hall, 2015*), but this strategy appeared, however, not suitable for larger ecosystems (*Sánchez-Carrillo & Álvarez Cobelas, 2017*). Waterfalls decrease the thickness of the boundary layer at the air/water interface, leading to massive gaseous exchanges with the atmosphere over short distances (*Chen et al., 2004*; *Teodoru et al., 2015*; *Leibowitz et al., 2017*). Hence, waterfalls induce $CO_2$-outgassing and associated shifts in DIC-$\delta^{13}$C values in acidic running waters where carbonate dissolution cannot compensate for the loss of $CO_2$ (*Palmer et al., 2001*; *Doctor et al., 2008*). Rapid degassing and equilibration to atmospheric values and associated shifts in DIC-$\delta^{13}$C below waterfalls should induce changes only in algal $\delta^{13}$C values, and theoretically not affect isotopic ratios of allochthonous organic matter. As algal production has been shown to be an important source of C in similar streams/rivers (see also *Rasmussen, 2010*), we expected a shift in scale dependent IFIs linked to a shift in algal $\delta^{13}$C values. Therefore, waterfall systems could provide an excellent natural experimental set-up to quantify impacts of changes in isotopic baselines on ecological inferences from IFIs in a range of rivers and streams varying in size.

The aim was to study impacts of changes in isotopic baselines on the evaluation of food web structure using IFIs, and we hypothesized that DIC isotopic shifts brought by $CO_2$-outgassing at waterfall sites should induce changes in food web structure inferences based on IFIs. Similarity in food web structures at above- versus below-waterfall sampling sites from five rivers and streams in Southern Quebec (Canada) was tested by comparing taxonomic composition and $\delta^{15}$N values to assess positioning of trophic guilds. We also compared IFIs (calculated in the $\delta^{13}$C- $\delta^{15}$N space) for invertebrate communities at above- and below-waterfall sites, and differences in IFIs within waterfall paired-sites were interpreted as a result of changes in algal isotopic baselines brought by waterfall-induced DIC isotopic shifts. We hypothesized that IFIs exhibit different sensitivities to changes in isotopic baselines due to calculation methods. IFIs calculated using the dispersion of species in a $\delta$-space (isotopic functional richness) are therefore scale-dependent and should be more sensitive to changes in isotopic baselines than those based on the distribution of species within a $\delta$-space (isotopic functional evenness and diversity).

## MATERIAL AND METHODS

### Site description and sampling protocol

Five waterfalls (with vertical drops ranging from 18 to 72 m), from small streams to large rivers (widths ranging from 6 to 50 m), were studied in Southern Quebec, Canada (between 46−47°N and 72−73°W). Their catchment areas are situated on the Canadian Shield (corresponding to a metamorphic geological formation), making the running water weakly conductive and slightly acidic (ranging from 20 to 50 $\mu$S cm$^{-1}$ with an average pH value around $6.3 \pm 0.4$ at investigated sites). To use changes in DIC-$\delta^{13}$C values brought by waterfall $CO_2$-outgassing as an integrative indicator of changes in algal isotopic baselines, each site was sampled at two locations immediately upstream and downstream of the waterfall (hereafter above- and below-waterfall), and the maximum distance between the two sampling points was 300 m. Paired sampling locations were also selected to have similar environmental conditions (water velocity, riverbed substrates, water depth, surrounding

vegetation cover, canopy cover, etc.). Therefore, as habitat structures in each waterfall paired-sites were similar, food web structures of above- and below-waterfall invertebrate communities were also expected to be similar.

## C-gas sampling and carbon stable isotope analysis of DIC

Water chemistry (partial pressure of $CO_2$: $pCO_2$, DIC-$\delta^{13}C$) was measured at each sampling site to characterize biogeochemical effects of waterfalls and quantify expected shifts in algal $\delta^{13}C$ values. The selected sites were visited between 1 to 2 times in spring and summer (in early May and late June 2016). $pCO_2$ was measured using the headspace method (*Campeau & Del Giorgio, 2014*). 30 mL of water sample was collected from approximately 10 cm below the water surface, using a 60 mL polypropylene syringe, and 30 mL of ambient air was added into the syringe to create a 1:1 ratio (ambient air: water sample). Then, the syringe was vigorously shaken for 1.5 min to equilibrate the gases in the water and air fractions. 30 mL of the headspace was then injected into a 40 mL glass vial, prefilled with saturated NaCl solution (360 g $L^{-1}$ at 20 °C), through a butyl rubber septum. A second needle was used to evacuate the excess of NaCl solution. Vials were kept inverted for storage, and headspaces were analysed using a Shimadzu GC-8A Gas Chromatograph with flame ionization detector at University of Quebec at Montreal (Montreal, Canada). $pCO_2$ in water samples was then retrocalculated using the headspace ratio, water temperature and ambient air concentrations of $CO_2$ at studied sites. $pCO_2$ measurements were performed in duplicates for each sampling site. Supersaturation ratios (SR) were also calculated by dividing the gas water partial pressure by the atmospheric $CO_2$ concentration.

At each sampling site, 500 mL of water sample was also collected in duplicates in early May and late June 2016 to analyse DIC-$\delta^{13}C$. Water samples were filtered at 0.2 μm using nitrocellulose membrane filters and stored for a maximum of 72 h in the dark at 4 °C until analysis. 150 μL of phosphoric acid ($H_3PO_4$; 85%) was added into 12.5 mL amber borosilicate vials to ensure that all DIC content in the water sample would be converted into $CO_2$. Then, vials were flushed using Helium during 10 min to ensure a full evacuation of ambient air. 4 mL of water sample was injected in He-flushed vials through the rubber septa using fine needles, and vials were equilibrated at 20 °C for 18 h. DIC-$\delta^{13}C$ was obtained using with a ThermoFinnigan Gas Bench II coupled to an Isotope Ratio Mass Spectrometer (IRMS), and results were expressed as the delta notation with Vienna Pee Dee Belemnite as the standard: $\delta^{13}C$ (‰) = ([$R_{sample}$/$R_{standard}$] − 1) ×1000; where R = $^{13}C/^{12}C$. Sample measurement replications from internal standards (C1 = −3.0‰, and C5 = −22.0‰) produced analytical errors ($1\sigma$) of ± 0.3‰ ($n = 17$).

## Invertebrate sampling and carbon stable isotope

Each sampling station was sampled in early July 2016, and benthic invertebrates were collected in riffle sections using a kick-net (0.1 $m^2$, 600 μm mesh size). Equal sampling effort was applied to each habitat type within above- and below-waterfall sites. Invertebrate specimens were sorted immediately in the field into taxonomic groups and transported in the dark at 4 °C back to the laboratory 4–8 h later to be frozen at −20 °C until analysis. A small isotopic deviation can be observed using this method (see also *Feuchtmayr &*

*Grey, 2003*; *Wolf et al., 2016*), but we assumed that this effect was the same for all samples. Invertebrates were identified at the genus level (*Merritt & Cummins, 1996*), and specimens were then classified into different feeding groups as herbivores, detritivores, and predators (*Thorp & Covich, 2009*; *Merritt & Cummins, 1996*; Electronic supplementary material S2). Samples were then dried at 60 °C for 72 h and ground into fine powder. Carbon ($\delta^{13}$C) and nitrogen ($\delta^{15}$N) stable isotopic ratios were then analysed using an Isotope Ratio Mass Spectrometer interfaced with an Elemental Analyser (EA-IRMS) at University of Quebec at Trois-Rivieres (Trois-Rivieres, Canada). Results were expressed according to the delta notation (see above). Sample measurement replications from three internal standards (STD67: $\delta^{13}$C $= -37$ ‰ and $\delta^{15}$N $= 8.6$‰, UTG40: $\delta^{13}$C $= -26.2$‰ and $\delta^{15}$N $= -4.5$‰, and BOB1 $\delta^{13}$C $= -27$‰ and $\delta^{15}$N $= 11.6$‰) produced analytical errors (1 $\sigma$) of $\pm 0.02$‰ for $\delta^{13}$C values and $\pm 0.2$‰ for $\delta^{15}$N values ($n = 148$).

## Isotopic functional indices and data analysis

Non-metric Multidimensional Scaling (NMDS) was used to visualize dissimilarities among/within invertebrate communities at waterfall sites, and the Bray–Curtis index was used to measure dissimilarities of invertebrate communities based on presence/absence data. T-tests were also performed on trophic guilds $\delta^{15}$N values to compare their trophic positions in food webs at above- and below-waterfall sites.

Means of $\delta^{13}$C and $\delta^{15}$N values of all individuals for each species calculated at each sampling site were used to derive six IFIs following *Layman et al. (2007)*: $\delta^{13}$C range (CR), $\delta^{15}$N range (NR), total area of the convex hull encompassing all the observations (TA), mean distance to centroid (CD), mean nearest neighbour distance (MNND) and standard deviation of nearest neighbour distance (SDNND). Layman's IFIs can be grouped into isotopic functional richness (CR, NR and TA); isotopic functional divergence (CD); and isotopic functional evenness (NND and SDNND). As isotopic functional richness indices (CR, NR and TA) provide a quantitative indication of the extent of the isotopic niche space of the entire community and are calculated using the dispersion of species in the $\delta$-space (*Layman et al., 2007*; *Jackson et al., 2011*), we hypothesized that those IFIs should be more sensitive than those based on the distribution of species in the $\delta$-space (CD, MNND and SDNND; Appendix. 1). All indices were calculated using *SIAR* package for R (*Parnell & Jackson, 2013*). Principal component analysis (PCA) was also performed to display changes in structural properties of above- and below-waterfall invertebrate communities and provide an overview of relationships between IFIs and changes in DIC- $\delta^{13}$C values. All statistical analyses and plots were performed using the R 3.5.2 software (*R Core Team, 2018*).

## RESULTS

A total of 36 water samples were analysed for $p$CO$_2$ and DIC- $\delta^{13}$C. In our study, sampled running waters were slightly acidic (with an average pH value around $6.3 \pm 0.4$) and carbonate dissolution cannot compensate for the loss of CO$_2$. Therefore, waterfalls induced consistent increase in DIC-$\delta^{13}$C values induced by rapid CO$_2$-outgassing (Fig. 1). Below-waterfall DIC-$\delta^{13}$C values were always higher than those of above-waterfall samples, and
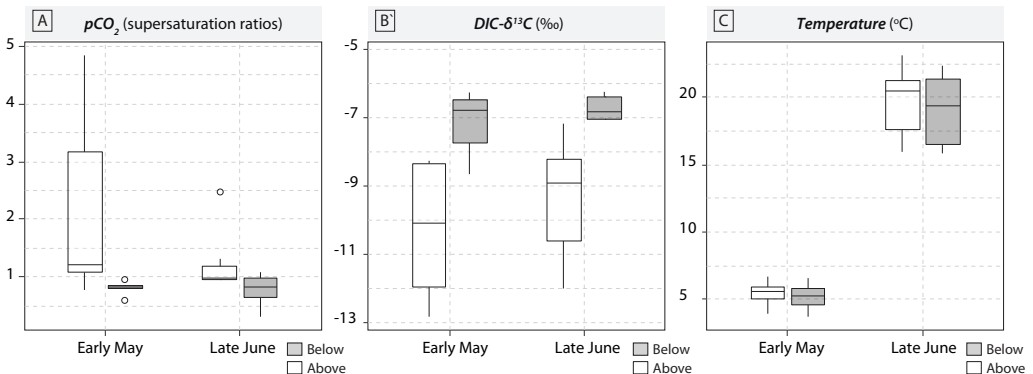

**Figure 1** Boxplots of $p$CO$_2$ concentrations in water samples (expressed in supersaturation ratio, SR) and DIC-$\delta^{13}$C values (expressed in ‰) for the five waterfall sites sampled in early May and late June 2016. Grey symbols represent below-waterfall samples, whereas white symbols correspond to above-waterfall sites.

results showed an average increase of 2.2‰ (ranging from $-3.8 \pm 0.2$‰ to $-0.9 \pm 0.2$‰; Fig. 1). Temporal comparisons between the two sampling periods (early May and late June 2016) revealed strong differences in $p$CO$_2$ and water temperature (rising from $5.3 \pm 1$ °C in May to $19.3 \pm 2.6$ °C in June), but smaller effects for DIC-$\delta^{13}$C values relative to the above- vs. below-waterfall sites (Fig. 1).

Most of the genera caught at each above-waterfall site were also found at below-waterfall sites (Table 1). Bray-Curtis index between each paired waterfall sites ranged 0.13–0.4, whereas the average Bray–Curtis index value calculated among sites was $0.38 \pm 0.09$, suggesting that compositional differences among communities were higher among waterfalls than within each paired-sites, and high similarity of invertebrate community composition of each paired waterfall sites was also further validated through an NMDS plot (Fig. 2). Furthermore, the $\delta^{15}$N values of invertebrate specimens ranged 0.8–10.2‰ (Fig. 3), and no significant difference was observed for each trophic guild between above- and below-waterfall samples ($p$-value > 0.05; Fig. 4). The $\delta^{13}$C values of invertebrate specimens ranged from $-33.3$‰ to $-23$‰ (Fig. 3) and showed consistent increases in $\delta^{13}$C values for below-waterfall samples. The $\delta^{13}$C-$\delta^{15}$N biplots visually highlighted strong similarities between above- and below-waterfall invertebrate communities, showing large overlaps in isotopic spaces encompassing all species locations (Fig. 3). Large differences in $\delta^{13}$C values were also observed among trophic guilds (Fig. 5).

Calculations of IFIs showed notable changes between above- and below-waterfall IFI values (Fig. 6). Isotopic functional evenness and divergence indices (SDNND, MNND and CD, respectively) showed relatively small changes between above- and below-waterfall sites (Fig. 6). In contrast, isotopic functional richness indices (mainly TA and especially CR, as expected, but also NR in a lesser extent) followed important changes between above- and below-waterfall sites, and with few exceptions isotopic functional richness indices were lower at below-waterfall sites than at above-waterfall samples (Fig. 6). Principal
**Table 1  Taxonomic list of macroinvertebrate collected in the sampling sites (1/0 refer presence/absence).** Specimens were classified into different functional groups according to their theoretical feeding behaviours: herbivore, detritivore, and predator (*Thorp and Covich, 1991*; *Merritt & Cummins, 1996*). Waterfall systems are abbreviated to the first four letters. A refers above-waterfall sites and B to below-waterfall sites.

| Order | Family | Genus | Trophic guild | Bost | | Bull | | Dorw | | Pour | | Ursu | |
|---|---|---|---|---|---|---|---|---|---|---|---|---|---|
| | | | | A | B | A | B | A | B | A | B | A | B |
| Crustacea | Cambaridae | *Orconectes* | Detritivore | 1 | 0 | 0 | 0 | 0 | 0 | 1 | 1 | 0 | 0 |
| Ephemeroptera | Ephemerellidae | *Serratella* | Detritivore | 0 | 1 | 1 | 0 | 0 | 1 | 0 | 0 | 1 | 0 |
| Plecoptera | Pteronarcyidae | *Pteronarcys* | Detritivore | 0 | 0 | 0 | 1 | 0 | 0 | 1 | 1 | 1 | 1 |
| Trichoptera | Hydropsychidae | *Hydropsyche* | Detritivore | 1 | 1 | 1 | 1 | 1 | 1 | 1 | 1 | 1 | 1 |
| Trichoptera | Hydropsychidae | *Macrostemum* | Detritivore | 1 | 1 | 0 | 0 | 1 | 1 | 0 | 1 | 1 | 1 |
| Trichoptera | Limnephilidae | *Pycnopsyche* | Detritivore | 0 | 0 | 1 | 0 | 0 | 0 | 0 | 0 | 1 | 0 |
| Coleoptera | Psepheridae | *Psephenus* | Herbivore | 0 | 0 | 0 | 0 | 1 | 0 | 0 | 0 | 0 | 1 |
| Ephemeroptera | Ephemerellidae | *Drunella* | Herbivore | 1 | 0 | 0 | 0 | 1 | 0 | 0 | 0 | 1 | 1 |
| Ephemeroptera | Heptageniidae | *Epeorus* | Herbivore | 1 | 1 | 1 | 1 | 1 | 1 | 1 | 1 | 1 | 1 |
| Ephemeroptera | Ephemeridae | *Ephemera* | Herbivore | 0 | 0 | 1 | 0 | 0 | 0 | 0 | 0 | 0 | 0 |
| Ephemeroptera | Oligoneuriidae | *Isonychia* | Herbivore | 1 | 0 | 0 | 0 | 1 | 1 | 0 | 0 | 1 | 1 |
| Megaloptera | Corylaridae | *Coryladus* | Predator | 0 | 1 | 1 | 0 | 1 | 1 | 1 | 1 | 1 | 1 |
| Odonata | Aeshnidae | *Boyeria* | Predator | 1 | 1 | 1 | 1 | 1 | 1 | 1 | 1 | 1 | 1 |
| Odonata | Cordulegastridae | *Cordulegaster* | Predator | 0 | 0 | 1 | 1 | 0 | 0 | 0 | 0 | 1 | 1 |
| Odonata | Gomphidae | *Hagenius* | Predator | 0 | 0 | 0 | 0 | 0 | 0 | 0 | 0 | 1 | 1 |
| Odonata | Gomphidae | *Ophiogomphus* | Predator | 1 | 0 | 1 | 1 | 0 | 0 | 1 | 0 | 0 | 1 |
| Plecoptera | Perlidae | *Acroneuria* | Predator | 1 | 1 | 1 | 1 | 1 | 1 | 1 | 0 | 1 | 1 |
| Plecoptera | Perlidae | *Claassenia* | Predator | 1 | 1 | 1 | 1 | 0 | 1 | 0 | 1 | 1 | 1 |
| Trichoptera | Philopotamidae | *Chimarra* | Predator | 1 | 0 | 1 | 0 | 1 | 1 | 0 | 0 | 0 | 0 |
| Trichoptera | Rhyacophilidae | *Rhyacophilia* | Predator | 0 | 1 | 0 | 1 | 0 | 0 | 1 | 1 | 1 | 1 |
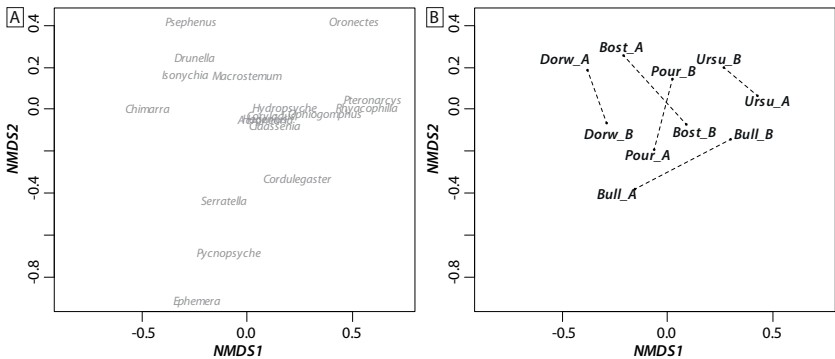

**Figure 2    NMDS ordination biplot (A: individual taxa; B: communities) of invertebrate communities at waterfall sites based on Bray_Curtis index (presence/absence).** Waterfall sites are abbreviated to the first four letters, and each pair of sampling site are linked with a dotted black line.

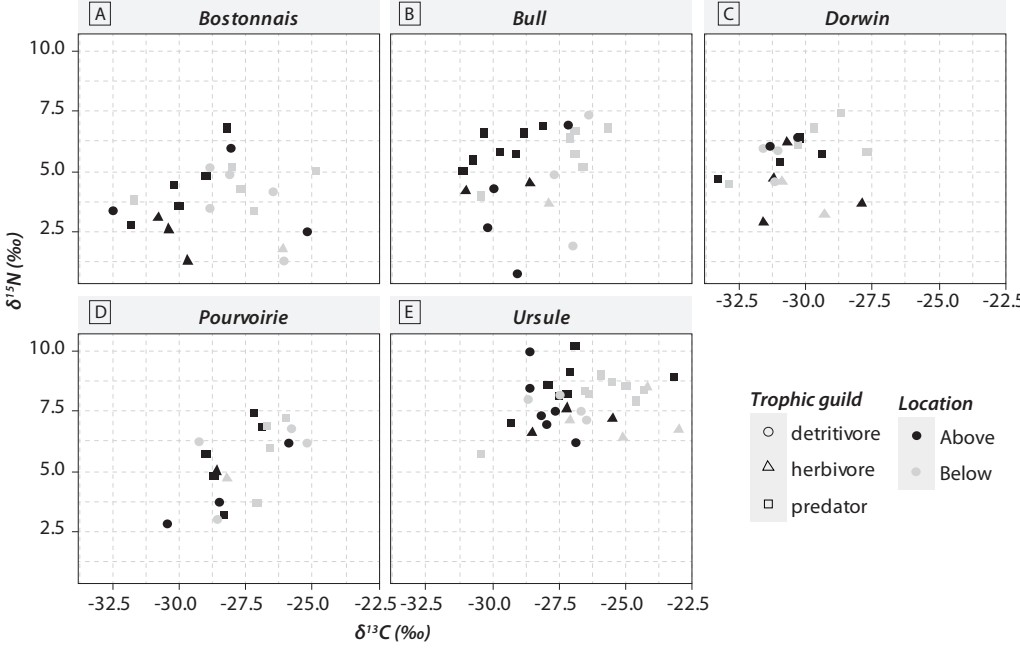

**Figure 3    Carbon ($\delta^{13}$C) and nitrogen stable isotopic ratios ($\delta^{15}$N) of invertebrate communities collected in the five waterfall sites.** (A) Bostonnais, (B) Bull, (C) Dorwin, (D) Pourvoirie, (E) Ursule. Each point on the graph represents the mean value of 2–7 individuals of each species. Error bars are omitted for simplicity.

component analysis (PCA) gave an overview of differences between above- and below-waterfall invertebrate communities and illustrated the overall relationships between all IFIs. The first two PCA axes explained 54.7% and 29.1% of the total variance, respectively (Fig. 7A). The first PCA axis mainly explained isotopic functional divergence and evenness indices (Fig. 7B), whereas the second PCA axis explained CR and NR (Fig. 7B). As expected, additional projection of DIC- $\delta^{13}$C values in the factorial map revealed visual correlation

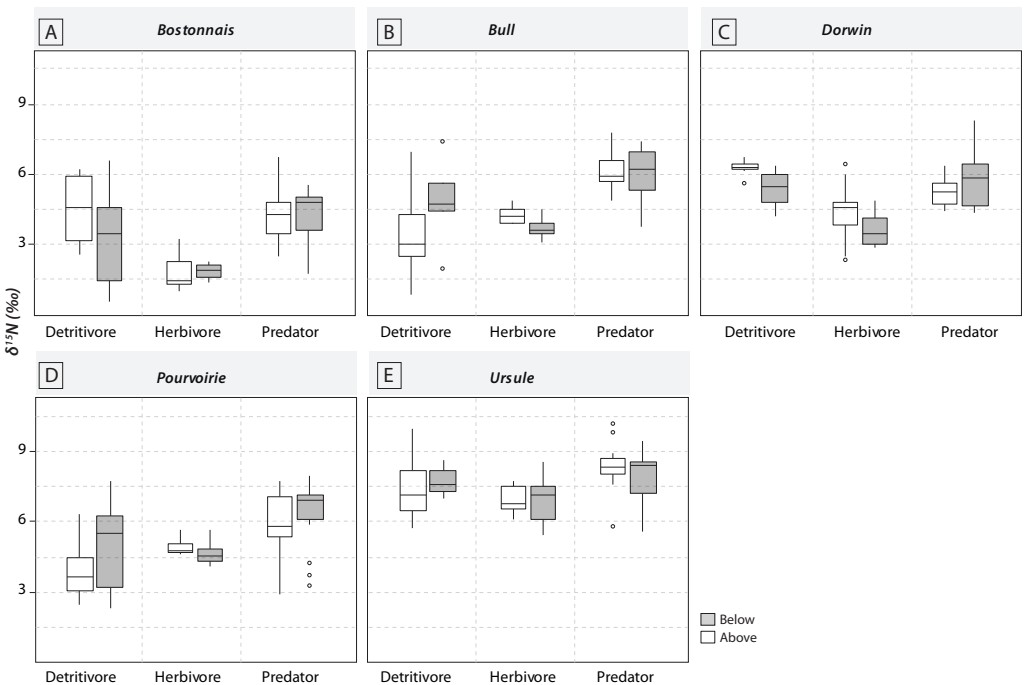

**Figure 4** **Boxplots of trophic guilds $\delta^{15}N$ values (expressed in ‰) for the five waterfall sites.** (A) Bostonnais, (B) Bull, (C) Dorwin, (D) Pourvoirie, (E) Ursule. Grey symbols represent below-waterfall samples, whereas white symbols correspond to above-waterfall sites. None of the relationships were significant ($t$-test; $p$-value $> 0.05$).

with CR (as $\delta^{13}C$ range of invertebrate specimens; Fig. 7B). Overall, PCA1 and PCA2 scores of below-waterfall invertebrate communities were higher than those of above-waterfall communities (Fig. 7A), suggesting that IFIs values were often lower at below-waterfall sampling sites than at above-waterfall locations.

## DISCUSSION

### Waterfalls, community structure and basal resources

NMDS (Fig. 2) and taxonomic list (Table 1) showed only little changes in taxonomic composition and suggested that compositional differences among communities were higher among waterfalls than within each paired site. Furthermore, no changes in $\delta^{15}N$ values of aquatic consumers (Fig. 4) were reported suggesting that the analyzed organisms occupied similar trophic positions in the food webs (*Cabana & Rasmussen, 1996*)) and might therefore feed on similar diets above and below the waterfall sites. Therefore, in our study, waterfalls did not significantly impact the food web structure of invertebrate communities, and these results could strengthen previous findings showing the absence of major effect of waterfall on invertebrate community composition in four tropical rivers (*Baker et al., 2016*).

Waterfall $CO_2$-outgassing and associated shift in DIC- $\delta^{13}C$ values should induce punctual changes in algal $\delta^{13}C$ values and could also help to decipher the respective

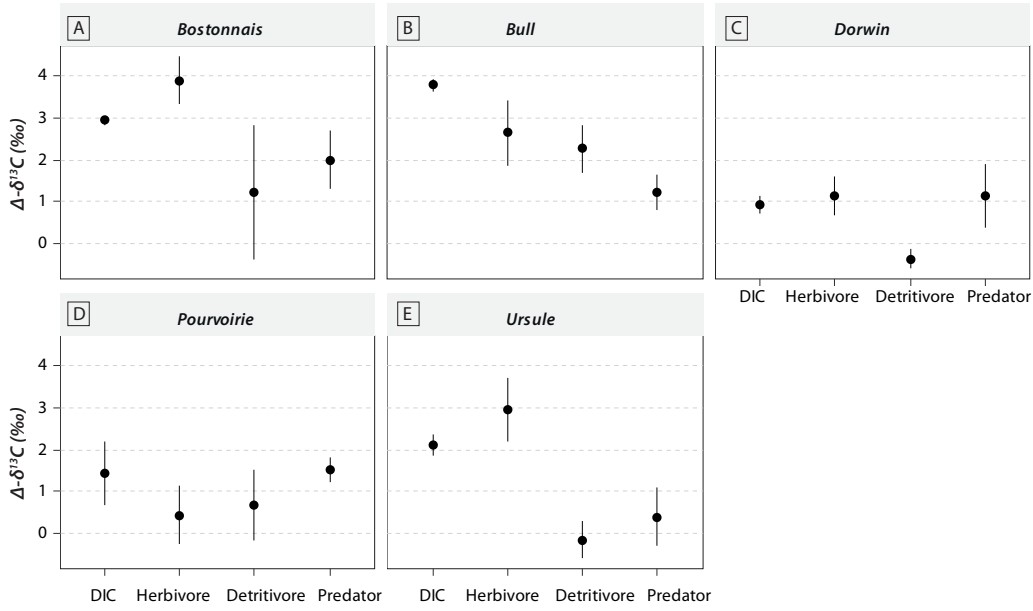

**Figure 5** Changes in $\delta^{13}$C values for DIC and consumers belonging to different trophic guilds between above- and below-waterfall samples (with $\Delta\delta^{13}C = \delta^{13}C_{below} - \delta^{13}C_{above}$). (A) Bostonnais, (B) Bull, (C) Dorwin, (D) Pourvoirie, (E) Ursule. Trophic guilds are abbreviated to the first four letters (e.g., Herbivore becomes "Herb").

contribution of allochthonous and autochthonous carbon to lotic food webs. With few exceptions, $\delta^{13}$C values of trophic guilds were higher at below-waterfall sites than those of above-waterfall samples (Fig. 3), supporting the view that aquatic invertebrates mainly feed on in-stream algae (*Tanentzap et al., 2017*). However, differences among trophic guilds observed in our study might also suggest varying reliance on algae (Fig. 5). Surprisingly, large isotopic shifts were also observed for detritivores (but were in general smaller than those for herbivores), suggesting an important dependence on autochthonous sources for these organisms theoretically relying on detritus (Fig. 5; *McNeely, Clinton & Erbe, 2006*). Therefore, our study could support the hypothesis of the prevalence of autochthony in river and stream food webs (*Brett et al., 2017*; *Tanentzap et al., 2017*) but could also emphasize the issue of trait plasticity for inveterate leading to differences between actual and theoretical feeding habits.

## Sensitivity of IFI to changes in isotopic baselines

IFIs have become increasingly used in aquatic ecology (*Olsson et al., 2009*; *Abrantes, Barnett & Bouillon, 2014*; *Dézerald et al., 2018*; *Burdon, McIntosh & Harding, 2020*), but IFI concept mainly relies on untested assumptions. In this study, we consider waterfall systems as a natural experimental set-up to quantify impacts of changes in isotopic baselines on ecological inferences from IFIs. We consider that waterfall $CO_2$-outgassing and associated shift in DIC- $\delta^{13}$C values should induce punctual changes in algal $\delta^{13}$C values and therefore help to understand how changes in isotopic baselines impact upon IFIs.

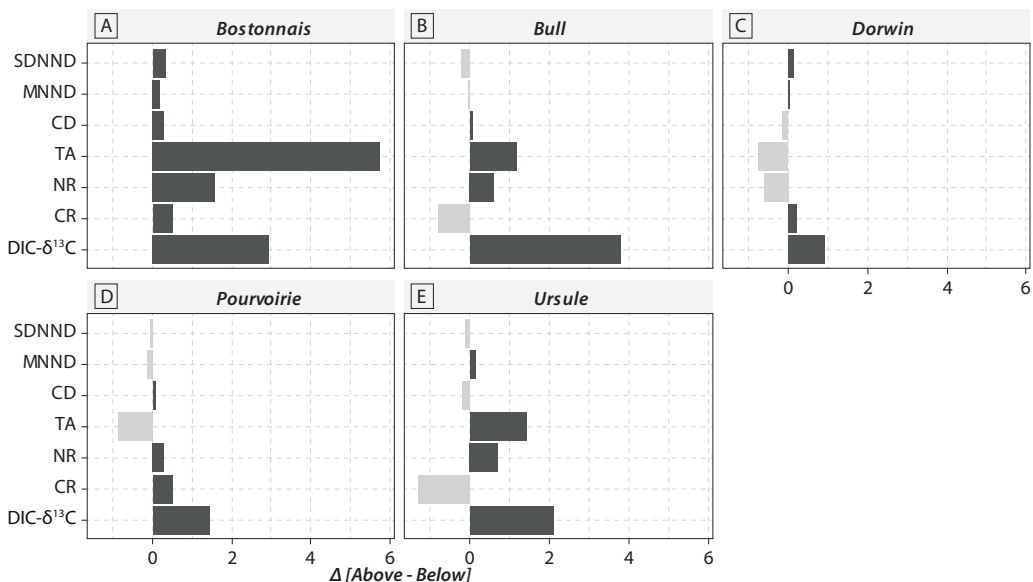

**Figure 6 Barplot comparing isotopic functional indices and DIC-$\delta^{13}$C values obtained from above-vs. below-waterfall sites of the five streams and rivers.** (A) Bostonnais, (B) Bull, (C) Dorwin, (D) Pourvoirie, (E) Ursule. Each barplot indicates the difference in each IFI value between above-vs. below-waterfall sites ($\Delta$ = above –below), and colours refer to the direction of changes (black = positive; grey = negative).

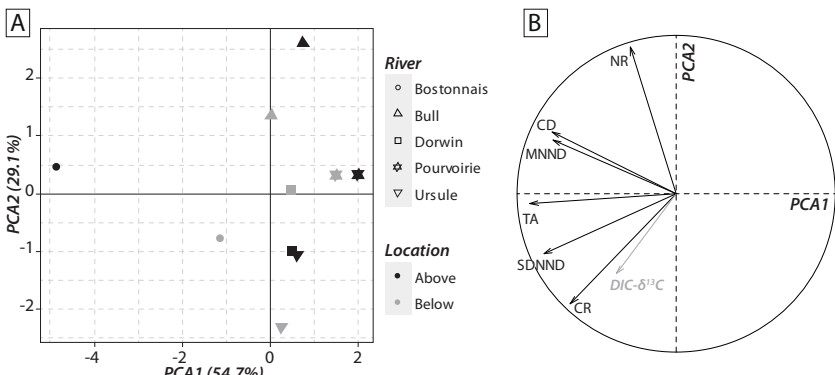

**Figure 7 (A) Factorial map of the principal component analysis (PCA) performed on isotopic functional indices (PCA2 versus PCA1). (B) Correlation circle representing isotopic functional indices' contributions to the first two axes of the PCA.** Symbols refer to rivers, whereas colours refer to sampling locations. DIC-$\delta^{13}$C (grey arrow) has been added to the correlation circle as a passive variable.

Notable differences in in the expected direction in IFI values were reported between above- and below-waterfall sites (Fig. 6), and changes were likely driven by shifts in DIC-$\delta^{13}$C values (Fig. 7B). Isotopic functional evenness and divergence indices (CD, SDNND and MNND) were only slightly impacted by changes in isotopic baselines (Fig. 6). Indeed, those indices were calculated based on the distribution of species in the $\delta$-space (*Layman et al., 2007*), and ecological inferences were therefore not strongly impacted by changes

in the vertical or horizontal distribution of specimens in the isotopic space. In contrast, isotopic functional richness indices (CR, NR and TA), calculated on species dispersion in the $\delta$-space and providing a quantitative indication of the extent of isotopic niche space of the entire community, were strongly influenced by changes in algal $\delta^{13}$C values (Figs. 6 and 7). Moreover, differences in CR, NR and TA values between above- and below-waterfall sites often exceeded the range of changes previously reported in literature (*Rigolet et al., 2015*). These results could strengthen previous findings that these indices are very sensitive to changes in ranges of consumer $\delta^{13}$C and $\delta^{15}$N values (*Brind'Amour & Dubois, 2013*; *Syväranta et al., 2013*). Hence, isotopic functional richness indices (CR, NR and TA) might be frequently misinterpreted in river studies comparing food webs across sites and/or over time with fluctuating isotopic baselines.

Our study suggested that the reliability of IFI inferences can be strengthened by identifying changes in IFIs driven by variability in isotopic baselines. In that vein, different propositions have already been made in the literature to improve the understanding of food web structure and resource partitioning in consumers (e.g., *Jabot et al., 2017*). The most promising approach likely consists of increasing the number of isotopes studied (like hydrogen and sulphur: *Doucett et al., 2007*; *Proulx & Hare, 2014*) and including other types of data (such as gut content, fatty acids contents or compound specific isotope analysis). Indeed, the combination of these complementary proxies will provide new insights on actual energy pathways through food webs. By enabling a better understanding of trophic interactions in food webs, future IFI-based studies will contribute to better document food web structural properties.

## CONCLUSION

Our study demonstrated that changes in isotopic baselines can impact the evaluation of river food web structure using IFIs, but these effects depended on IFI types (i.e., being higher for IFIs measuring species distribution in the $\delta$-space than for other IFIs), leading to potential misinterpretations of IFIs in river studies where isotopic baselines generally show high temporal and spatial variabilities. The identification of isotopic baselines and their associated variability, and the use of independent trophic tracers to identify the actual energy pathways through food webs must be a prerequisite to IFIs-based studies to strengthen the reliability of ecological inferences of food web structural properties.

## ACKNOWLEDGEMENTS

Special thanks are addressed to Dany Bouchard (UQTR, Canada) for technical assistance during stable isotopes analyses. We also thank Micheline Bertrand and Pauline Jeanneret for their help during fieldwork.

### Funding

Financial support was provided by the Conseil Régional de Franche-Comté and the National Research Council of Canada. The funders had no role in study design, data collection and analysis, decision to publish, or preparation of the manuscript.

### Grant Disclosures

The following grant information was disclosed by the authors:
The Conseil Régional de Franche-Comté.
The National Research Council of Canada.

### Competing Interests

The authors declare there are no competing interests.

### Author Contributions

- Simon Belle conceived and designed the experiments, performed the experiments, analyzed the data, prepared figures and/or tables, authored or reviewed drafts of the paper, and approved the final draft.
- Gilbert Cabana conceived and designed the experiments, performed the experiments, authored or reviewed drafts of the paper, and approved the final draft.

### Field Study Permissions

The following information was supplied relating to field study approvals (i.e., approving body and any reference numbers):

No field permit required as we only accessed public land.

### Data Availability

The raw measurements are available in the Supplemental Files.

### Supplemental Information

Supplemental information for this article can be found online at http://dx.doi.org/10.7717/peerj.9999#supplemental-information.

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
