# Peer review of "Effects of changes in isotopic baselines on the evaluation of food web structure using isotopic functional indices"

_PeerJ, doi:10.7717/peerj.9999_

## Round 0.1 · original submission · Major Revisions

The work has been submitted for three reviews. All colleagues agree that some corrections must be made prior to the publication of the paper. Please take into account each of the comments and resend the paper for consideration.

Reviewer 1 ·

Basic reporting

The manuscript by Belle and Cabana entitled “Effects of changes in isotopic baselines on the evaluation of food web structure using isotopic functional indices” tested the validity of Layman’s IFIs in natural streams. The results indicated that some indices strongly depend on isotopic baselines, but some were not. The information is useful in applying “isotope space” study in food web research.

Experimental design

The use of waterfall as a controlled experiment in natural streams is a good idea. The experimental design is simple and straightforward.
One uncertain point is the authors sampled DIC in early May and late June 2016, but stream invertebrates in early June. As the environmental condition seems to change drastically from snow melting season to summer in Canada, the growth of each living organism seems to be high. Concentrations and isotope ratios of DIC are instantaneous, and short-lived algae may change their isotope ratio rapidly, but long-lived predator may integrate the signals through time. Do the author consider that the isotope ratios of DIC and invertebrate communities are comparable as in Fig. 5 and 6?

Validity of the findings

The final conclusion is clear. I agree with the authors that we need to test various isotopic indices to compare for relevant types of questions.

Additional comments

Minor comments below
L. 120 the same environmental conditions: Could you add simple descriptions among sites? Readers may want to make use of similar settings.
L. 152 delta-definition: The factor 1000 is an extraneous numerical factor and should be deleted (Coplen2011 Rapid Commun. Mass Spectrom).
L. 153 and other standards, too. Internal standards (C1 = -3 ‰, and C5 = -22 ‰): We don’t know each value itself, but the values should be written with significant figures, like -3.0 ‰.
L. 158 benthic invertebrates were sampled: Exact where did the authors sample invertebrates? Riffles, pools, or mixed?
L. 162 A small isotopic deviation can be observed using this technique: Indicate the reasons why the methods may cause the deviation.
L. 190 Remove “calculate”.
L. 205 “increase of -2.2 ‰” should be “increase of 2.2 ‰”.
L. 212 displayed high similarity: How did the authors identify “high similarity”?
L. 221: (Fig. 6): Fig. 5 comes later. Thus Fig. 6 should change to Fig. 5.
L. 271 (Fig. 4): Fig. 6?
L. 272 (Fig. 5.B): (Fig. 7.B)?
L. 284 and/or over time with fluctuating isotopic baselines: This seems true, but the authors didn’t test the hypothesis (see 2. Experimental design).

Table 1 Explain “1” and “0”, also “A” and “B”.
Fig. 5 Trophic guilds are abbreviated to the first four letters (e.g., Herbivore becomes “Herb”): but not abbreviated.

Reviewer 2 ·

Basic reporting

no comment

Experimental design

no comment

Validity of the findings

no comment

Additional comments

It seems that this manuscript is the revised version I have reviewed previously in another journal. The content has been improved considerably with newly added analyses such as NMDS, PCA, and IFIs, all of which were not shown in the previous manuscript. The results suggest that these indices are not very much sensitive to environmental change between above- and below-water falls. I think that the authors did a good job to wrap up their research, except for one critical issue as follows.

The main aim (Lines 93-94) and conclusion (Lines 286-288) of this study should be reframed, because both autochthonous and allochthonous baseline data (e.g., algae and terrestrial plants) were unfortunately not present. I strongly disagree to use DIC as baseline, because there is isotopic fractionation against 13C between DIC and algae. Its size and variation are both greater than the delta13C difference in DIC between above- and below-water falls. However, this criticism might be avoidable if the authors used herbivores and detritivores as aquatic- and terrestrial-baselines, respectively (Post 2002 Ecology). This is advantageous over using benthic algae as baseline, which show a large variation in delta13C depending on environmental heterogeneity such as water velocity (Finlay et al. 1999 L&O) and on DI13C variation (Lines 67-70).

Reviewer 3 ·

Basic reporting

This study performed the analysis of isotopic functional indices (IFIs) for a natural experimental set-up to quantify impacts of changes in algal isotopic baselines on ecological inference. This study was well conducted and very interesting for isotope ecology. However I have few minor comments.

For NMDS, please describe NMDS stress for the Figure 2.

Figure 4 I could not see x and y ticks.

Figure 5 What the error bar means here?

Experimental design

no comment

Validity of the findings

no comment

---

## Round 0.2 · accepted · Accept

Thank you very much for including all the changes suggested by the reviewers. I am happy to inform you that your paper is ready to be published.

Reviewer 2 ·

Basic reporting

no comment

Experimental design

no comment

Validity of the findings

no comment

Additional comments

Thank you for your revision. I think the manuscript has been improved substantially.